# A lipid gating mechanism for the channel-forming O antigen ABC transporter

Christopher A. Caffalette[1], Robin A. Corey[2], Mark S.P. Sansom [2], Phillip J. Stansfeld[2] & Jochen Zimmer[1]

Extracellular glycan biosynthesis is a widespread microbial protection mechanism. In Gram-negative bacteria, the O antigen polysaccharide represents the variable region of outer membrane lipopolysaccharides. Fully assembled lipid-linked O antigens are translocated across the inner membrane by the WzmWzt ABC transporter for ligation to the lipopoly-saccharide core, with the transporter forming a continuous transmembrane channel in a nucleotide-free state. Here, we report its structure in an ATP-bound conformation. Large structural changes within the nucleotide-binding and transmembrane regions push conserved hydrophobic residues at the substrate entry site towards the periplasm and provide a model for polysaccharide translocation. With ATP bound, the transporter forms a large trans-membrane channel with openings toward the membrane and periplasm. The channel's periplasmic exit is sealed by detergent molecules that block solvent permeation. Molecular dynamics simulation data suggest that, in a biological membrane, lipid molecules occupy this periplasmic exit and prevent water flux in the transporter's resting state.

[1] Molecular Physiology and Biological Physics, University of Virginia School of Medicine, Charlottesville, VA 22908, USA. [2] Department of Biochemistry, University of Oxford, Oxford OX1 3QU, UK. Correspondence and requests for materials should be addressed to J.Z. (email: jochen_zimmer@virginia.edu) or to P.J.S. (email: phillip.stansfeld@bioch.ox.ac.uk)

An abundant defense mechanism of Gram-negative pathogens against the host innate immune response is the exposure of lipid-linked complex carbohydrates on their cell surfaces[1–3]. These sugary coats establish extended barriers around the cell, reducing the risk of complement-mediated killing, regulating host-pathogen interactions, and protecting against desiccation[4]. O antigens represent one class of glycolipid conjugates and form the variable region of the lipopolysaccharide (LPS) molecules in the outer membrane (OM) of Gram-negative bacteria[5]. The O antigen polymers extend the conserved LPS core, which comprises lipid-A and the inner and outer core oligosaccharides. O antigens consist of serotype-defining oligosaccharide units of 3–5 sugars that are repeated multiple times to form hypervariable polysaccharides up to ~100 sugars long[5,6].

O antigens are assembled by two fundamentally different but equally abundant mechanisms. One pathway assembles the polymer from individual undecaprenyl pyrophosphate (UndPP)-linked repeat units on the periplasmic side of the inner membrane (IM)[5]. In the ABC transporter-dependent pathway, however, the full-length and also UndPP-linked O antigen is first assembled on the cytosolic side of the IM, and then translocated to the periplasmic side by an ABC transporter for ligation to the LPS core[5,7]. For both pathways, complete LPS molecules are shuttled to the OM via the Lpt pathway, while the UndPP is recycled via BacA[8,9].

ABC transporter-mediated translocation of UndPP-linked O antigens requires (1) the recognition of the substrate at the IM's cytosolic side, (2) the reorientation of the UndPP moiety and its lateral release into the periplasmic membrane leaflet, as well as (3) the translocation of the polysaccharide chain across the membrane. In an extended conformation, about 8–10 sugars of a linear polysaccharide suffice to span the average width of a lipid bilayer[10]. Because O antigens can exceed this number many times[11,12], their membrane translocation requires the formation of a continuous polysaccharide channel and likely a processive translocation mechanism.

The recent structure of the *Aquifex aeolicus* (Aa) WzmWzt O antigen ABC transporter in a nucleotide-free conformation provided the first insights into its translocation mechanism[13]. Firstly, in accordance with its function in biopolymer translocation, the transporter forms a continuous transmembrane (TM) channel sufficiently wide to accommodate a polysaccharide chain. Secondly, the transporter's Wzt nucleotide-binding domain (NBD) contains a unique α-helical extension near the cytosolic water-lipid interface. Because this 'gate helix' is also found in ABC transporters of Gram-positive bacteria that translocate UndPP-linked teichoic acids[14], it has been speculated that it is required for substrate recognition. Thirdly, the TM channel formed by Wzm is lined with aromatic residues that likely interact with the translocating polysaccharide via CH-π stacking interactions[15].

While the O antigen polymer may prevent conductance of water and small molecules during translocation, how the ABC transporter pushes the polysaccharide into its channel and how this channel closes in a resting state to maintain the membrane's permeability barrier remain important unresolved questions. To address these points, we determined the high-resolution crystal structure of AaWzmWzt in an ATP-bound conformation. The structure reveals the close association of the transporter's NBDs that coordinate two ATP molecules. Compared to the nucleotide-free state, the transporter undergoes large rigid-body movements of its nucleotide-binding and TM domains. Furthermore, TM helix 1 and a conserved loop at the putative substrate entry site bend towards the channel entrance and likely account for polysaccharide translocation. These conformational rearrangements in the ATP-bound state result in a continuous TM channel with two lateral exits towards the periplasmic membrane leaflet. The channel's periplasmic opening is accessible to detergent molecules

that form a hydrophobic plug and prevent solvent molecules from spanning the entire channel. Molecular dynamics (MD) simulations of the membrane-embedded transporter confirm that, in a biological membrane, lipid molecules readily occupy the channel's periplasmic opening after substrate release and suggest their crucial role in transporter gating.

## Results

**Structure of the ATP-bound O antigen transporter**. To gain mechanistic insights into O antigen polysaccharide translocation, we stabilized and crystallized a catalytically inactive AaWzmWzt ABC transporter in an ATP-bound conformation. ATP hydrolysis was prevented by introducing a single point mutation in the transporter's Walker B motif, replacing Glu167 with glutamine (referred to as AaWzmWzt$_{EQ}$)[16]. In addition, the transporter's carbohydrate-binding domain (CBD) at the C terminus of its NBD was removed to facilitate crystallization, as described previously[13]. The CBD extensions are found in a subset of O antigen ABC transporters and are likely involved in recruiting the substrate to the transporter and initiating translocation[17]. CBD-truncated ABC transporters are fully functional in vivo if the CBD is co-expressed as a separate domain[18].

In its ATP-bound state, the AaWzmWzt$_{EQ}$ transporter formed well-diffracting crystals, allowing structure determination at one of the highest reported resolutions for an ABC transporter (Table 1). The ATP-bound AaWzmWzt$_{EQ}$ structure was determined by molecular replacement at a resolution of 2.05 Å and includes residues 1–256 of Wzm and 2–235 of Wzt. The transporter contains one ATP molecule per active site. Our high-resolution structure

**Table 1 Data collection and refinement statistics (molecular replacement)**

|  | AaWzmWzt$_{EQ}$ |
|---|---|
| **Data collection** | |
| Space group | P6$_5$22 |
| *Cell dimensions* | |
| a, b, c (Å) | 144.75, 144.75, 201.47 |
| α, β, γ (°) | 90, 90, 120 |
| Resolution (Å)[a] | 25–2.05 (2.09–2.05)[b] |
| $R_{pim}$ | 0.030 (0.495) |
| CC$_{1/2}$[c] | 0.998 (0.582) |
| I / σI | 14.6 (1.7) |
| Completeness (%) | 99.9 (99.9) |
| Redundancy | 12.1 (9.8) |
| **Refinement** | |
| Resolution (Å) | 25–2.05 |
| No. reflections | 78248 |
| $R_{work}$ / $R_{free}$ (%) | 16.4/18.2 |
| *No. atoms* | |
| Protein | 4230 |
| ATP | 31 |
| LDAO | 256 |
| All other non-protein/solvent | 94 |
| Water | 371 |
| *B-factors (Å$^2$)* | |
| Protein | 49.1 |
| ATP | 29.2 |
| LDAO | 81.1 |
| All other non-protein/solvent | 77.6 |
| Water | 64.2 |
| *R.m.s. deviations* | |
| Bond lengths (Å) | 0.008 |
| Bond angles (°) | 0.970 |

[a]Data collected on a single crystal
[b]Values in parentheses are for highest-resolution shell
[c]Correlation between intensities from random half-data sets[63]

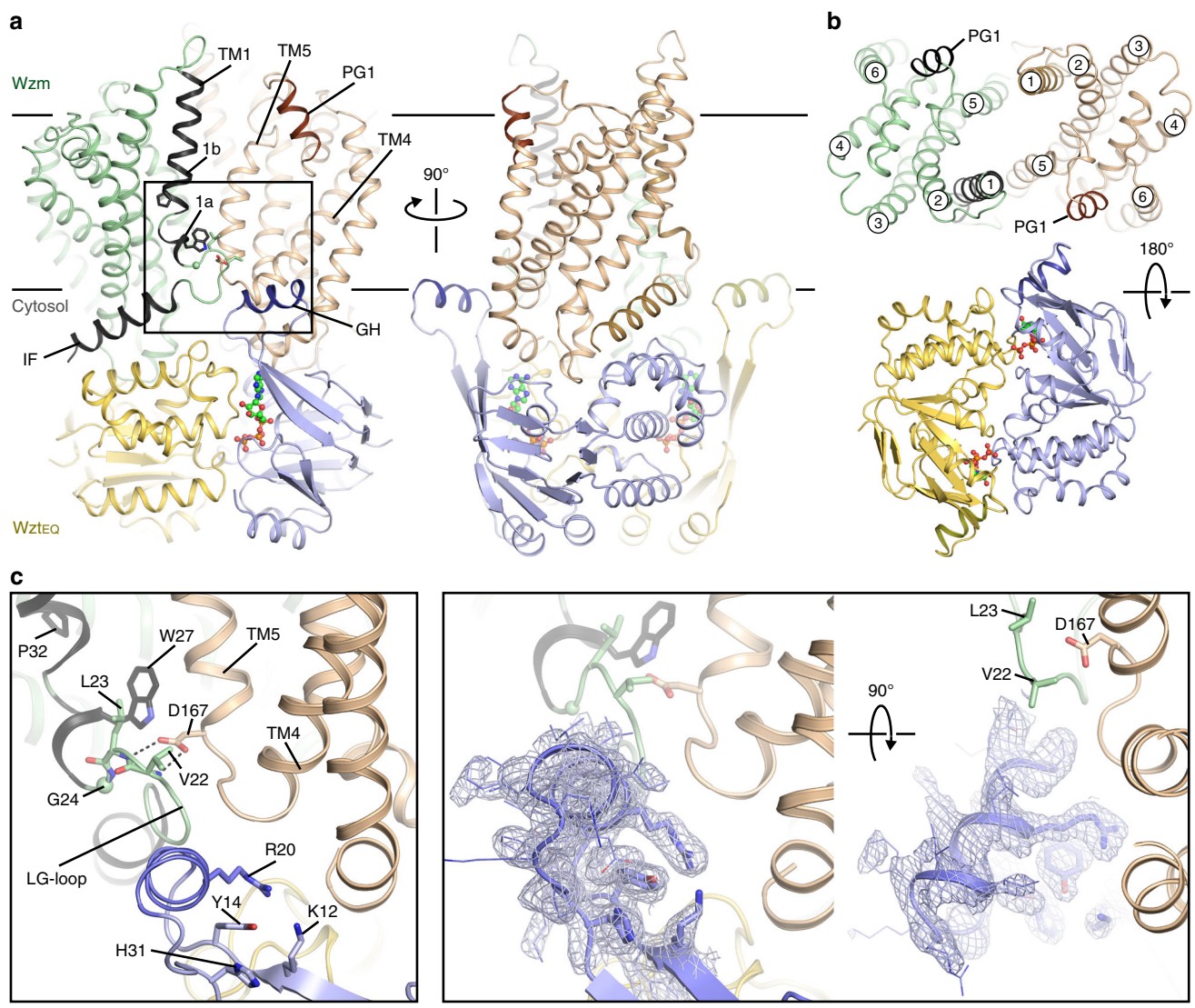

**Fig. 1** ATP-bound structure of the O antigen ABC transporter. **a** Two orthogonal views of AaWzmWzt_EQ with ATP bound. The Wzm protomers are shown in green and wheat with the N-terminal interface helix (IF) and TM helix 1 highlighted in black. The Wzt nucleotide-binding domains are shown in yellow and blue with bound ATP portrayed as ball-and-sticks. An extended loop of Wzt contains the cytoplasmic gate helix (GH, purple) that sits at the Wzm protomer interface. IF and TM1 are connected by the LG-loop. Lines show approximate lipid bilayer boundaries. **b** Periplasmic and cytosolic views of AaWzmWzt. Each Wzm protomer contains six transmembrane helices, and the loop between Wzm TM5/6 forms a short periplasmic gate helix (PG1). **c** The gate helix forms an electropositive pocket near the Wzm interface. Left: Interaction of the GH and LG-loop with the Wzm protomers. Right: Two orthogonal views of the GH with a 2Fo-Fc composite omit electron density map contoured at 1σ shown as a gray mesh

confirms the previously determined lower resolution nucleotide-free architecture of AaWzmWzt[13], with the exception that Wzm's N-terminal interface helix (IF helix, amino acids 3–19) was out of register by one residue.

ATP binding stabilizes the closed conformation of the transporter's NBDs (Fig. 1)[19]. Although crystallized in the presence of ATP and magnesium chloride, we observe that the nucleotide's phosphate groups are only coordinated by water molecules and protein. The transporter binds ATP at its NBD using conserved Walker A and B, H-loop, and signature motifs that have been previously described (Fig. 1 and Supplementary Fig. 1)[19,20]. The signature motif in O antigen and related ABC transporters consists of the sequence $Y_{142}$-S-S/T-G-M-X-X-R/K-L-A/G-$F_{152}$, of which Ser143 contacts ATP's γ-phosphate and Met146 packs into a hydrophobic pocket formed by Tyr142 and Phe134 (Supplementary Fig. 1).

Within the membrane-spanning segment, the Wzm subunits contact each other only via TM helices 1 and 5, similar to the

nucleotide-free state[13]. The N-terminal IF helix rests at the interface of the nucleotide-binding and TM domains and is connected to TM helix 1 via an extended loop that packs against the connection of TM helices 4 and 5 of the opposing subunit (Fig. 1). The extended loop (hereafter referred to as 'LG-loop') ends with a conserved $L_{23}$-G motif and is stabilized through several conserved interactions. These include side chain contacts between Leu23 and Trp27 within the same subunit and backbone contacts of Val22 and Leu23 with the carboxylate of the conserved Asp167 in the opposing subunit (Fig. 1c). Past the LG-loop and only observed in the ATP-bound conformation, TM helix 1 is divided by the conserved Pro32 into a short $3_{10}$-helix (TM1a, residues 25–28) and TM1b (residue 30–47) (Fig. 1).

The transporter's cytosolic gate helix, formed by an extended connection of β-strands 1 and 2 of the NBD, packs with its N-terminus against the TM4–5 loop of the same half-transporter as well as the LG-loop of the opposing AaWzmWzt_EQ heterodimer (Fig. 1c). At this position, the helix forms a positively charged

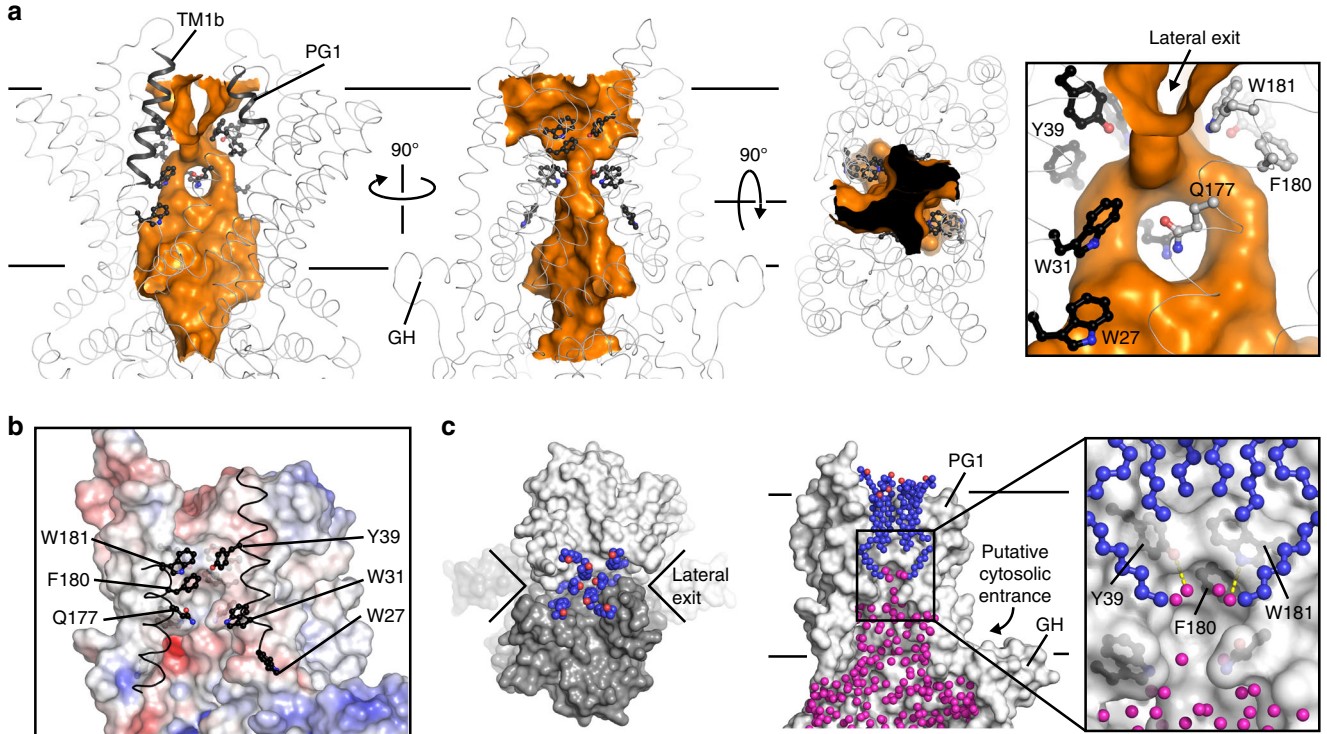

**Fig. 2** WzmWzt forms a large transmembrane channel in the ATP-bound conformation. **a** Surface representation of the AaWzmWzt$_{EQ}$ channel calculated with a 2.5 Å-radius probe shown as an orange surface. Right panel, zoomed in view of the channel near the periplasmic lateral exit. Aromatic residues lining the channel are shown as black and gray ball-and-sticks for different Wzm protomers. The transporter backbone is shown as a gray ribbon. **b** Electrostatic surface potential of the channel calculated with the APBS plugin in PyMOL (scale −5.0 (red) to +5.0 (blue) kT/e). TM1 and 5 of the opposing Wzm protomer are shown as a black ribbon with selected aromatic residues in ball-and-sticks. **c** LDAO molecules occupy the periplasmic opening of the channel. Left: Periplasmic view of AaWzmWzt$_{EQ}$. Right: The AaWzmWzt$_{EQ}$ channel is shown with one WzmWzt half transporter removed. Crystallographically resolved water molecules inside the Wzm channel are shown as magenta spheres. AaWzmWzt$_{EQ}$ is shown as a gray surface and LDAO molecules as blue and red spheres

pocket at its interface with the transmembrane domain (TMD) containing the hydroxyl group of the conserved Tyr14 at its center (Fig. 1c). An electropositive nature of this pocket is conserved among O antigen and teichoic acid transporters and established in AaWzt by the side chains of Lys12, Arg20, and His31 that point straight into it (Fig. 1c).

It has previously been speculated that the cytosolic gate helix forms the substrate-binding site for recruitment of UndPP-linked O antigens or teichoic acids[13]. Indeed, tyrosine residues surrounded by positively charged side chains are frequently found in undecaprenyl-phosphate and UndPP binding pockets, as exemplified by the enzymes aminoarabinose[21], farnesyl[22], and oligosaccharyl transferase[23], among others. In these cases, the tyrosine's hydroxyl together with arginine and lysine side chains contact the substrate's phosphate group, while the lipid moiety resides in a hydrophobic pocket: a similar coordination is likely in AaWzmWzt (Fig. 1).

**ATP-bound WzmWzt forms a continuous TM channel.** In a nucleotide-free conformation, AaWzmWzt forms a TM channel wide enough to accommodate a translocating polysaccharide chain[13]. Strikingly, this TM channel is maintained in the ATP-bound conformation, albeit with a slightly smaller diameter (Fig. 2). Using a 2.5 Å radius probe, a continuous solvent-accessible pore is identified starting at the cytosolic water-lipid interface near the gate helix and ending on the periplasmic side with a large funnel-shaped opening (Fig. 2a). The periplasmic opening contains lateral exits towards the periplasmic bilayer

leaflet formed at the Wzm protomer interface between TM helix 1 and the periplasmic gate helix (PG1) (Fig. 2a). Of note, a rescue mutation that restores ATPase activity of the related *Staphylococcus aureus* wall teichoic acid transporter in the presence of a small molecule inhibitor localizes to this periplasmic lateral exit[24]. This suggests that conformational changes of the TMDs and lateral exit opening are coupled to ATP hydrolysis.

The channel radius is widest within the cytosolic membrane leaflet, constricts to about 2.5 Å at its midpoint, and widens significantly towards the periplasm and laterally towards the periplasmic lipid leaflet. Atomistic MD simulations in a 1-palmitoyl-2-oleoyl-sn-glycero-3-phosphoethanolamine (POPE) lipid bilayer confirm that the ATP-bound transporter architecture is very stable with a constant pore profile over the course of a 100 ns simulation (Supplementary Fig. 2). The constriction halfway across the membrane is formed by Gln177 of TM helix 5, which points towards the channel lumen and interacts with its symmetry mate in the opposing half transporter (Fig. 2a). A different side chain orientation of Gln177 would eliminate this constriction.

As observed in the nucleotide-free conformation, the Wzm channel has mainly hydrophobic character with Trp27, Trp31, Tyr39, Gln177, Phe180, and Trp181 forming putative contact points with the translocating O antigen. In particular, Tyr39, Phe180, and Trp181 encircle the channel in an 'aromatic belt' forming a hydrophobic ring that may regulate polymer and solvent permeation (see below) (Fig. 2a, b).

Interestingly, the experimental electron density reveals six complete and four partially ordered detergent molecules, assigned

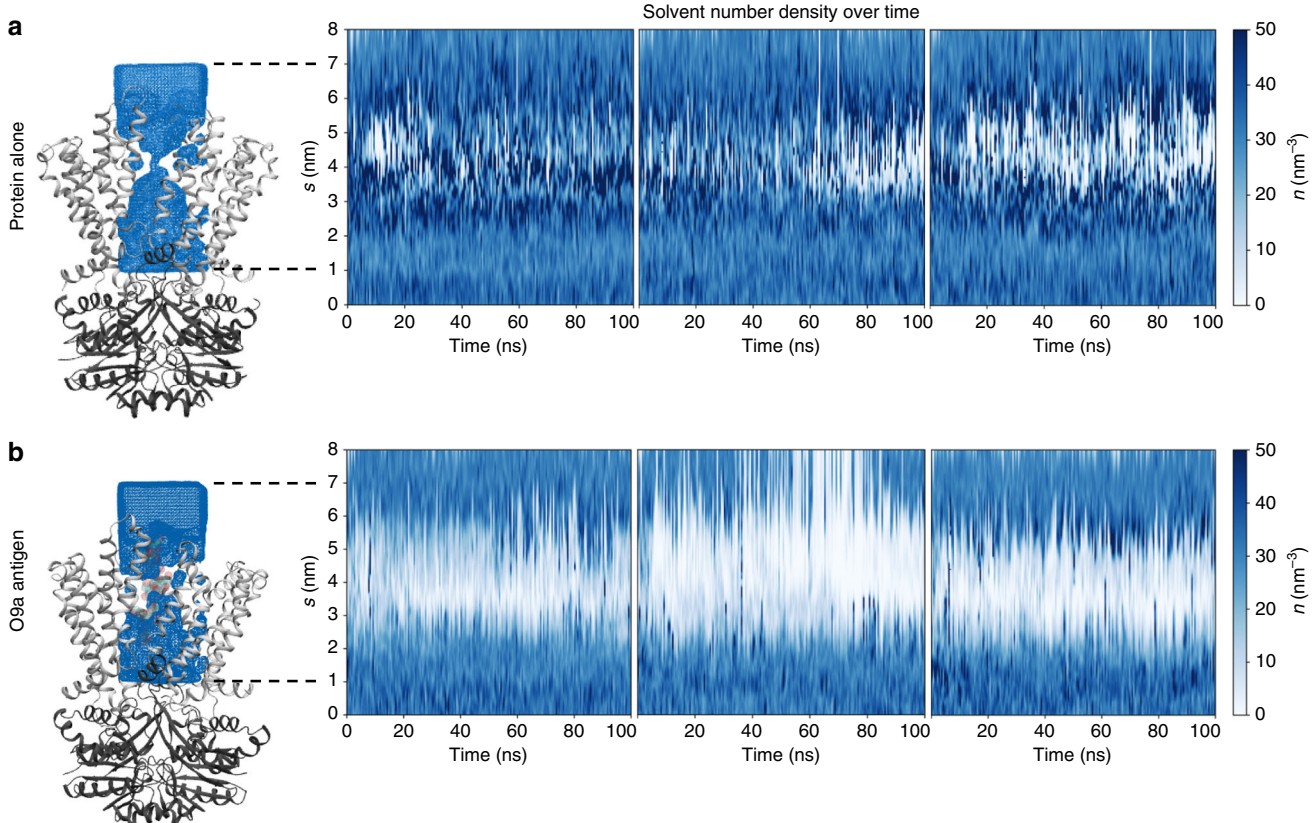

**Fig. 3** Water flux through WzmWzt is prevented by substrate. **a** Snapshot of AaWzmWzt following 100 ns atomistic simulation, showing only the protein (Wzm in light gray, Wzt in dark gray) and a small region of water density around Wzm (blue). A column of water can be seen through the center of Wzm. Right: Quantification of the water solvent density, $n$, across the 100 ns simulation. The $Y$-axis, $s$, represents the position along the channel axis. Data from three repeats are shown. **b** Substrate-engaged transporter, with the sugars of the substrate shown as green and red transparent spheres. Right: Quantification of the water solvent density plotted the same as in **a**

as lauryl-N,N-dimethylamine-N-oxide (LDAO) molecules, within the periplasmic channel opening and lateral exit, respectively (Fig. 2c and Supplementary Fig. 3). Most of the interactions with the transporter occur through TM helices 1 and 5, which form the seam between the Wzm protomers. In this position, the detergent molecules occupy the entire channel volume, thereby excluding solvent. Indeed, crystallographically resolved water molecules inside the TM channel are only observed within the first (cytosolic) section of the channel up to the aforementioned aromatic belt and are blocked from reaching the periplasm by the detergent molecules (Fig. 2c).

The transporter's TM channel is sufficiently wide to accommodate a polysaccharide chain. Indeed, MD simulations with a modeled UndPP-linked *E. coli* O9a repeat unit[25] inside the channel require no significant changes in pore dimensions (Supplementary Fig. 4). In this model, the oligosaccharide resides within the channel's periplasmic half, and the undecaprenyl moiety extends into the bilayer through the lateral exit. Channel lining residues, including Tyr39, Phe180, and Trp181 of the aromatic belt pack against the O antigen and/or hydrogen bond with its hydroxyl groups (Supplementary Fig. 4c). Combined, the observed channel properties suggest that our structure represents WzmWzt's ATP-bound conformation during polysaccharide translocation, requiring no further channel widening to accommodate a polysaccharide (Supplementary Figs. 2 and 4).

**Lipids and substrate control water flux through the TM channel.** Like all membrane transporters, WzmWzt needs to

carefully gate its TM channel; the pore should be sufficiently wide to permit substrate passage only, with no space for additional water flux. To probe the effects of substrate binding on the permeation of water through the Wzm channel, we performed a series of atomistic MD simulations on the O9a antigen-bound transporter and analyzed water flux through its pore (Fig. 3). Both the conformation of the docked O antigen and that of the transporter were stable over 100 ns, with extensive contacts made between them as outlined above (Supplementary Fig. 4).

Analysis of water occupancies using VMD and the CHAP package[26–28] reveals a clear solvent-excluded volume in the channel near its aromatic belt in the presence of the substrate, while in its absence, this volume is filled with water molecules (Fig. 3). Therefore, the translocating substrate appears to prohibit water permeation through the channel.

This raises the question of how the transporter maintains the membrane's permeability barrier in the absence of a polysaccharide chain. To test this, we ran simulations of the channel in a coarse-grained (CG) representation[29,30], bound to a hepta-mannose oligosaccharide and embedded in a POPE bilayer (Fig. 4a). The oligosaccharide was then slowly pulled laterally from the channel over a period of 4 μs. In each of these simulations, before the oligosaccharide leaves the channel, a lipid molecule tightly associates with and enters the transporter through the lateral exit (Fig. 4a and Supplementary Movie 1). As the oligosaccharide leaves, the lipid molecule replaces it inside the channel (Fig. 4a) such that the pore is constantly occluded. Indeed, MD simulations of the lipid-bound state reveal that water permeation is blocked, comparable to the O antigen inserted state

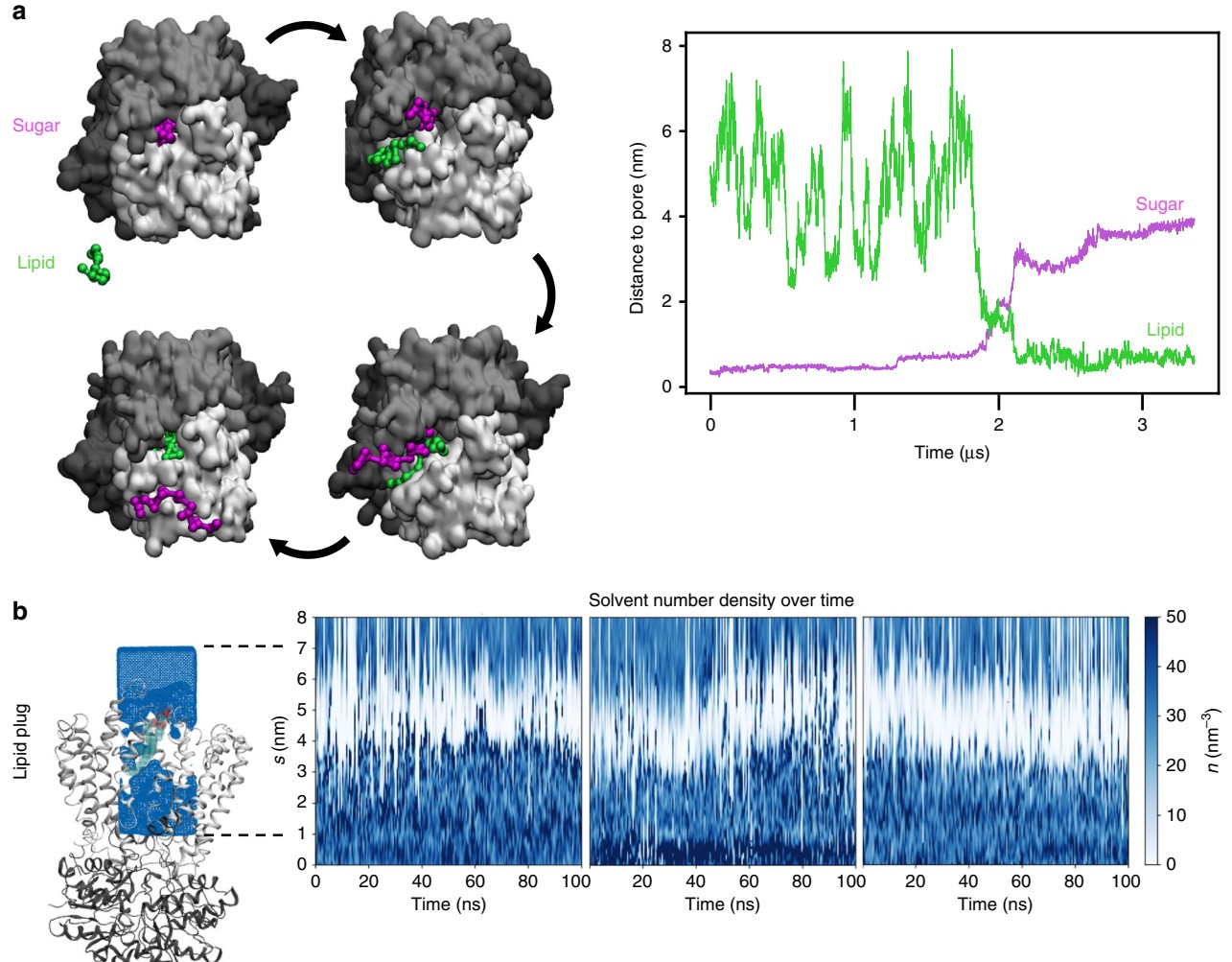

**Fig. 4** Lipids gate the WzmWzt channel. **a** Details of the CG pulling simulations, with snapshots of a representative system viewed from the periplasm. Wzt is shown as black surface, and the two Wzm protomers are shown as light gray and white surfaces. The sugar molecule, which is pulled from the channel, is shown in pink and the replacing POPE molecule in green. The rest of the membrane is omitted for clarity. Right: Quantification of sugar–lipid swap for a representative simulation. **b** Post-100ns atomistic simulation snapshot of the lipid-engaged AaWzmWzt transporter and quantification of the water solvent density over the simulation, as per Fig. 3. The lipid molecule is shown as green and red transparent spheres

(Fig. 4b). This suggests the presence of a 'lipid-plug' to seal the channel in the absence of substrate.

**ATP-induced movements suggest a stepwise transport mechanism.** The introduced Wzt Walker B mutation allowed trapping of the transporter in an ATP-bound state, revealing a closed conformation of its NBDs, as usually observed in nucleotide-bound ABC transporters (Figs. 1 and 5 and Supplementary Fig. 1). Upon ATP binding, each NBD rotates by about 10° and moves by about 9 Å toward the other, thereby closing the ATP binding sites at their interface (Supplementary Movie 2). Within each NBD, the ATP-induced rotation of the RecA-like subdomain toward the helical region moves the cytoplasmic gate helix by about 8 Å towards the channel entrance (Fig. 5 and Supplementary Fig. 5)[31,32]. This closure is stabilized by aromatic stacking interactions of the conserved Tyr11 with ATP's adenine group (Supplementary Figs. 1 and 5a).

The inward movement of the gate helix is possible due to a similar displacement of the TMDs. The Wzm protomers slide against each other in the plane of the bilayer, while maintaining their interactions between TM helices 1 and 5. Overall, TM helices 2 to 6 move as a rigid body with only minor structural changes compared to the transporter's nucleotide-free state (Fig. 5b and Supplementary Fig. 5b). TM helix 1, however, shifts significantly by ~10 Å towards the channel axis primarily by sliding along TM helices 2 and 3 (Supplementary Fig. 5b). This transition could be induced by the inward movement of the NBD's gate helix and facilitated by the bending of TM helix 1 around Pro32 (see above). During these transitions, the LG-loop remains associated with the invariant Asp167 of TM helix 5 in the opposing half transporter (Fig. 5a). Because this interaction persists, the LG-loop flips from a position parallel to the water-lipid interface in the absence of nucleotide to perpendicular to it in the ATP-bound conformation (Fig. 5a).

Combined, the ATP-induced conformational changes push the gate helix, LG-loop, and TM helix 1 towards the periplasm. Relative to a reference point at the channel center near the periplasmic exit, the N terminus of the cytosolic gate helix and the conserved Leu23 of the LG-loop move ~4 Å and 9 Å towards this point, respectively, while His45 at the periplasmic end of TM helix 1 translocates ~2 Å away from it (Fig. 5a and Supplementary Movie 2). Assuming a direct interaction of these motifs with the translocating polysaccharide chain, the observed conformational changes likely account for polymer translocation.

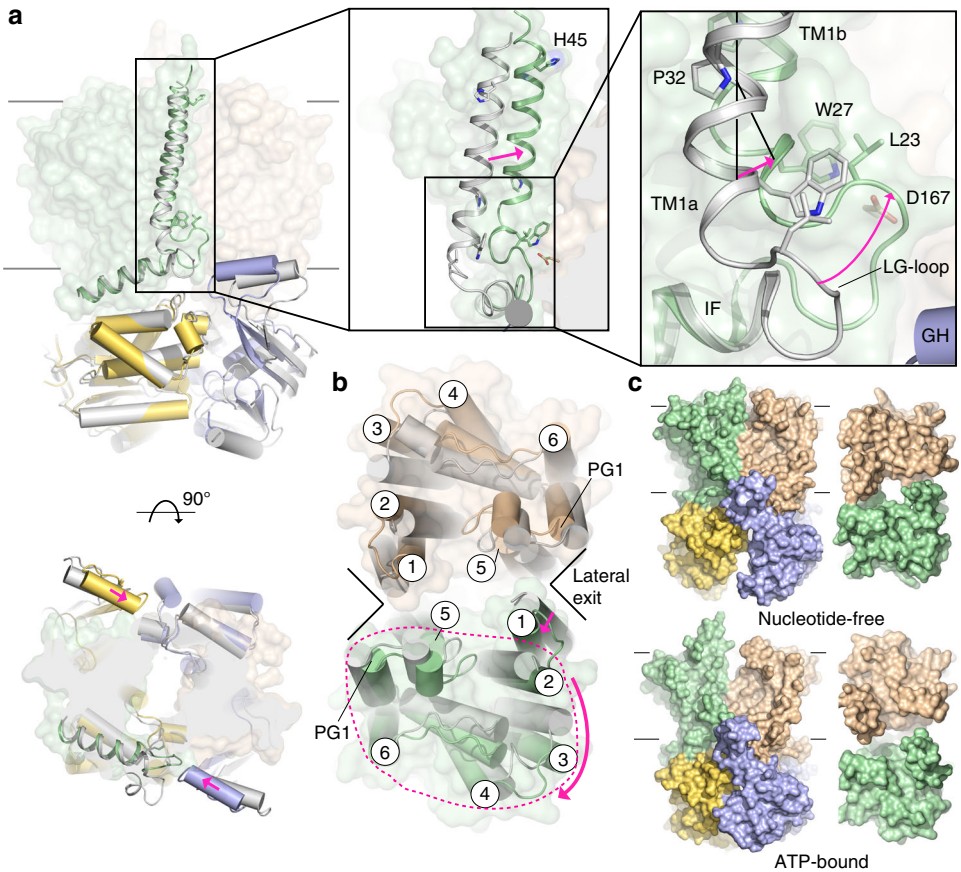

**Fig. 5** ATP-induced conformational changes of WzmWzt. **a** The Wzm and Wzt subunits of the nucleotide-free state (shown in gray) were individually structurally aligned with the corresponding subunits in the ATP-bound conformation. For Wzt, alignment was based on the helical subdomain (residues 82–192) and for Wzm, on TM helices 2–6 by secondary structure matching in Coot[52]. **b** Global alignment based on the NBD-dimer of the nucleotide-free (gray) and ATP-bound states. Shown is a periplasmic view of the Wzm protomers. **c** Surface representations of the nucleotide-free and ATP-bound WzmWzt conformations. In all panels, Wzt and Wzm are shown in yellow/blue and green/wheat, respectively

In the full-length AaWzmWzt transporter, the NBD's C terminus is attached to a CBD (Supplementary Fig. 6a). CBDs are found in ABC transporters that translocate 'capped' O antigens, which are modified at their non-reducing termini after completion of O antigen biosynthesis[7,25,33]. The CBDs specifically recognize these modifications. Crystallographic analysis of isolated CBDs from *E. coli* and *R. terrigena* Wzts confirm their jelly-roll like fold and dimeric organization[17,18], and the isolated AaWzt-CBD dimer interacts with the truncated transporter[13]. The interaction surface with the transporter likely also contains the binding site for the O antigen cap[18].

Because the distance between the NBDs' C termini remains between 40–48 Å in the nucleotide-free and ATP bound states (Supplementary Fig. 6b), we speculate that the CBDs do not restrict the movements of the NBDs during catalysis, yet conformational changes and functional properties could be modulated upon binding of the O antigen cap.

## Discussion

O antigen biosynthesis is a multistep process that results in the exposure of glycoconjugates on the bacterial cell surface. Biosynthesis relies on either the polymerization of lipid-linked O antigen repeat units in the periplasm or the orchestrated activities of membrane-localized cytosolic glycosyltransferases[5]. When fully assembled intracellularly, a high molecular weight polymer has to be transported to the IM's periplasmic side where both biosynthesis pathways converge. Here, the WaaL ligase covalently links the polysaccharide to the LPS core[5].

At a minimum, membrane transport of biological polymers requires substrate recognition on the membrane's cytosolic side, initiation of transport and insertion of the polymer into its transporter, as well as polymer translocation through a TM channel. For lipid conjugated polymers, including O antigens and teichoic acids, the lipid moiety also has to reorient in the membrane during this process, such that the lipid anchor moves from the cytosolic to the periplasmic membrane leaflet.

For O antigens and teichoic acids, the lipid anchor is undecaprenyl-phosphate, which is modified with an acetylated aminosugar (in most cases GlcNAc), thereby generating an UndPP-GlcNAc anchor[6]. Despite the absence of a substrate molecule, the AaWzmWzt architecture and, in particular, the location and physical properties of its cytosolic gate helix strongly argue that the substrate-binding site is located at the interface of the TMD with the gate helix (Figs. 1 and 6). At this location, the substrate would directly interact with the LG-loop at the TMD protomer interface, which in turn is likely to facilitate the insertion of the lipid head group into the transporter.

As substrate insertion occurs at the interface of the TMDs, with TM helices 1 and 5 forming the contacts (Fig. 1), the lipid anchor could reorient with its undecaprenyl moiety remaining in the bilayer and its charged head group in the transporter proper, as proposed for 'flipping' of UndPP-linked oligosaccharides by PglK[34]. Flexibility of the side chain packing at this interface likely forms a 'soft seam' to allow the passage of the undecaprenyl chain. Further, the distance over which the polyprenyl group has to be 'dragged' through the channel wall is significantly shortened

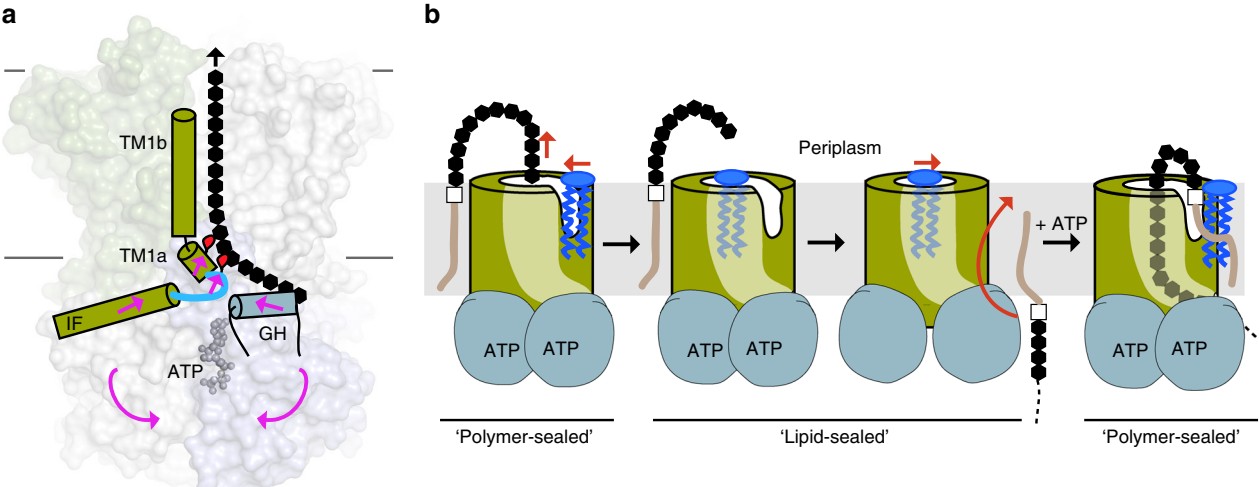

**Fig. 6** Model of O antigen membrane translocation. **a** Upon ATP binding and NBD closure, the cytosolic gate helix (GH) pushes against the LG loop connecting Wzm's IF and TM1. This movement correlates with the bending of TM1 and upward movement of TM1a. Conserved hydrophobic residues (Leu23 and Trp27, red ovals) interact with and push the polysaccharide chain into the transmembrane channel. **b** Membrane lipids gate the WzmWzt channel. Concomitant with the final translocation step that releases the polysaccharide into the periplasm, lipid molecules diffuse into the channel to form a hydrophobic plug. This plug is replaced by the substrate upon insertion of a new lipid-linked O antigen into the channel. The substrate's lipid moiety is released into the periplasmic membrane leaflet through the lateral exit, which also allows lipid diffusion as a gating mechanism. The substrate is shown as a brown line connected to a white square for UndPP and black hexagons for the O antigen

by the channel's lateral exit within the periplasmic bilayer leaflet (Fig. 6). The negative membrane potential likely contributes to the reorientation of the negatively charged UndPP head-group to the periplasmic side, while also preventing its backsliding, from the periplasmic to the cytosolic side.

The insertion of the substrate into the channel near the cytosolic gate helix would place the polysaccharide chain in direct contact with the LG-loop and the following TM helix 1a. The ATP-induced flipping motion of this loop and bending of TM1a are likely important transitions that mediate polysaccharide translocation. Both, the conserved Leu23 and Trp27 within these motifs could translocate the polysaccharide based on steric interactions.

Similarly, polypeptide translocation by hexameric AAA-ATPases, such as ClpX, Vps4, and Hsp104, also relies on steric interferences between bulky (usually aromatic) residues of a moveable loop and the translocation substrate[35–37]. In comparison, translocation through cellulose synthase is mediated by the movement of a short α-helix that interacts with the polysaccharide, thereby transporting the polymer one sugar unit at a time[38]. Our data suggest that O antigen ABC transporters use a similar translocation mechanism, combining a loop movement and helical bending motion (Fig. 6a). The distance over which the LG-loop and TM1a move are consistent with a processive translocation of 1–2 sugar units per ATP-induced conformational transition; similar step-sizes have been observed for cellulose synthase and ClpX[35,38].

While our data provide a plausible mechanism for the stepwise forward translocation of the polysaccharide, how the transporter resets between translocation steps without retro-translocating the substrate remains unknown. Possible mechanisms could rely on kinetic differences of active translocation versus Brownian diffusion and/or the presence of a unidirectional ratchet based on steric interactions. Yet, these detailed mechanistic analyzes must await structural information on substrate-bound WzmWzt states.

ATP-energized processive polymer translocation requires the opening and closing of the transporter's NBDs while the polymer resides in its TM channel, i.e. NBD closure should not collapse the TM channel. Our MD simulation data demonstrate that the

translocating polysaccharide prevents formation of a proton wire, crucial for maintaining the membrane's proton motive force during translocation. Yet, this permeability barrier has to be maintained in the absence of a translocating substrate. Instead of adopting a channel-free closed conformation, WzmWzt likely accomplishes this task by allowing lipid molecules to occupy its periplasmic channel opening in the absence of a substrate, such that membrane lipids, instead of protein domains, gate the TM channel (Fig. 6b).

Our MD simulation data suggest that a single POPE molecule suffices to gate the Wzm channel, although the periplasmic funnel is wide enough to accommodate multiple LDAO molecules. It is likely that the extended lipid acyl chains together with Wzm's aromatic belt form a sufficiently hydrophobic seal to prevent water permeation. This is supported by the observation that, in our crystal structure, solvent does not permeate past the most deeply inserted two detergent molecules near the channel's aromatic belt.

A lipid gating mechanism is possible because their diffusion likely exceeds the O antigen translocation rate by several orders of magnitude, ensuring that the transporter's channel is always occupied by either lipid molecules or the translocation substrate, but not solvent. Similarly, upon initiation of O antigen translocation, the UndPP head group could displace the lipid plug. Furthermore, as the lateral exit only opens in the ATP-bound state, lipid release into the bilayer can only occur during or after the ATP-induced power stroke. Thus, on substrate binding, the UndPP moiety likely reorients while displacing the lipid plug, and the O antigen inserts into the channel for processive translocation and prevention of water permeation.

Other microbial glycolipids implicated in pathogen survival are capsular polysaccharides, of which groups 2 and 3 are also secreted by ABC transporters[39]. Here, the transporter partners with periplasmic and OM components to form a continuous conduit across the cell envelope, similar to Type-1 secretion and multi-drug extrusion systems[40–42]. How these transporters select their phospholipid-linked substrates[43] and maintain the membrane's permeability barrier in a resting state are important unresolved questions.

Type-2 ABC exporters frequently reorient hydrophobic molecules within lipid bilayers[44–46]. The observed ATP-induced conformational changes of WzmWzt, in particular of its TMD, may thus represent a general mechanism by which these ubiquitous transporters operate.

## Methods

**Cloning, protein expression, and purification.** The AaWzmWzt$_{EQ}$ mutant used for crystallization was generated using the QuikChange method on an existing pETDuet wild type AaWzmWzt expression plasmid. Expression and purification of the mutant construct was carried out as previously described with certain exceptions as noted below[13]. AaWzmWzt$_{EQ}$ was overexpressed in *E. coli* C43(DE3) cells grown at 37 °C in lysogeny broth (LB) medium containing 100 µg mL$^{-1}$ ampicillin. Upon reaching an optical density at 600 nm (OD$_{600}$) of 0.6, expression was induced by the addition of 0.5 mM isopropyl-β-d-thiogalactoside (IPTG) and cells were incubated at 37 °C for 4 h until harvest by centrifugation.

Cell pellets were resuspended in RB buffer containing 20 mM Tris HCl pH 7.5, 100 mM NaCl, and 5 mM β-mercaptoethanol (β-ME) and lysed in a microfluidizer. The membrane fraction was pelleted by ultracentrifugation at 200,000 × *g* for 60 min in a Beckman Coulter Type 45 Ti rotor. Membrane pellets were then solubilized for 60 min at 4 °C in SB buffer containing 50 mM sodium phosphate pH 7.2, 100 mM NaCl, 20 mM imidazole, 5 mM β-ME, and 2% (w/v) polyoxyethylene (8)-dodecyl ether (C12E8, Anatrace). Remaining insoluble material was cleared by ultracentrifugation at 200,000 × *g* for 30 min in a Beckman Coulter Type 45 Ti rotor, and the supernatant was batch incubated with Ni-NTA agarose resin (ThermoFisher Scientific) for 60 min at 4 °C. Next, the Ni-NTA resin was packed into a gravity flow chromatography column and washed successively with three buffers: 50 mL WB1 buffer (RB buffer containing 22 mM imidazole and 5 mM LDAO); 50 mL WB2 buffer (RB buffer containing 40 mM imidazole and 5 mM LDAO); and 50 mL WB3 buffer (RB buffer containing 1.5 M NaCl, 22 mM imidazole and 5 mM LDAO). After washing, AaWzmWzt$_{EQ}$ was eluted in 35 mL EB buffer (RB containing 300 mM imidazole and 5 mM LDAO).

The eluate was concentrated to 10 mL using 50 kDa cut-off centrifugal filters (Amicon) and dialyzed overnight against a buffer containing 20 mM Tris pH 7.5, 100 mM NaCl, 5 mM magnesium chloride, 5 mM β-ME, and 5 mM LDAO. The following day, the dialysed protein was purified over a S200 gel filtration column (GE Healthcare) equilibrated with GF buffer containing 20 mM Tris HCl pH 7.5, 100 mM NaCl, 5 mM magnesium chloride, 0.5 mM tris(2-carboxyethyl)phosphine (TCEP), and 5 mM LDAO. The main peak fraction was concentrated to a 15 mg mL$^{-1}$ final concentration using a 50-kDa cut-off centrifugal filter (Amicon).

**Crystallization.** Prior to crystallization, the AaWzmWzt$_{EQ}$ transporter was combined 1:1 with GF buffer containing 4 mM ATP to achieve final concentrations of 7.5 mg mL$^{-1}$, 2 mM and 5 mM for AaWzmWzt$_{EQ}$, ATP and magnesium, respectively. Protein samples containing ATP and magnesium were incubated on ice for a minimum of 1 h before setting up crystal trays. Crystals were grown by sitting-drop vapor diffusion at 17 °C after combining 1 µL of well solution (21–25% v/v PEG 200, 0.02 M sodium acetate pH 4.0, 0.05–0.2 M ammonium sulfate, and 0.02 M sodium chloride) and 1 µL of protein solution. Crystals grew to their final size within 4 days and were harvested at 7 days. The crystals were cryoprotected by incremental addition of glycerol to the drop to a final concentration of 20%. Crystals were then harvested and flash cooled in liquid nitrogen prior to data collection.

**Structure determination.** Diffraction data for crystals belonging to space group P6$_5$22 were collected at the AMX beamline at Brookhaven National Laboratories (NSLS-II). Data were integrated in XDS[47] and reduced in Aimless using the CCP4 program suite[48] yielding a 2.05 Å dataset with one half-transporter in the asymmetric unit (Table 1).

The AaWzmWzt$_{EQ}$ structure was solved by molecular replacement using the transmembrane (Wzm) and nucleotide-binding domains (Wzt) of the AaWzmWzt nucleotide-free structure (PDB 6AN7) as search model ensembles with the program Phaser[49]. An initial round of rigid-body refinement was performed in Refmac5 as part of the CCP4 program suite. All subsequent refinement was performed with Phenix.refine[50] using TLS parameters[51], and the model was manually completed using Coot[52]. Although strong densities for ATP and LDAO molecules were observed early in refinement, these ligands were modeled at late stages of refinement. The structure was refined to R$_{work}$/R$_{free}$ values of 16.4 and 18.2% with 98.0 and 0.2% of residues in the favored and disallowed regions of the Ramachandran diagram, respectively. Coordinates and structure factors for AaWzmWzt$_{EQ}$ have been deposited in the Protein Data Bank. Figures were generated using PyMOL[53] and Wzm channel dimensions were analyzed using HOLLOW[54].

**MD simulations.** Docking substrate: A model substrate was created comprising a single repeat of the O9a antigen attached to UndPP (molecule was Me-P-3-αMan-(1–2)-αMan-(1–2)-αMan-(1–3)-αMan-(1–3)-αMan-(1–3)-αMan-(1–3)-αMan-(1–3)-βGlcNAc-UndPP). This model substrate was docked into the ATP-bound

AaWzmWzt$_{EQ}$ structure using AutoDock Vina[55], with the best scoring pose favoring a periplasmic positioning of the UndPP pyrophosphate chosen (Supplementary Fig. 4).

Atomistic MD simulation: All simulations were run using GROMACS 5.1.2[56]. Simulations were built using the AaWzmWzt$_{EQ}$ ATP-bound coordinates, and described using the Charmm36 force field[57]. Where used, all atom O9a antigen parameters were generated using CHARMM-GUI[58]. The protein-membrane structures were built into POPE membranes with explicit TIP3P water and sodium and chloride ions to a neutral charge and concentration of 0.15 M. The systems were energy minimized using the steepest descents method over 5000 steps, then equilibrated with positional restraints on heavy atoms for 1 ns in the NPT ensemble at 310 K with the V-rescale thermostat and semi-isotropic Parrinello-Rahman pressure coupling[59,60]. Production simulations were run without positional restraints with 2 fs time steps over 100 ns. Systems were analyzed for pore size and water flux using the CHAP package[27,28].

Coarse-grained (CG) simulation: Using the three independent post-100 ns substrate-bound simulations as a starting point, CG simulations were run using Martini 2.2[29,30] of the AaWzmWzt$_{EQ}$ complex in a POPE membrane bound to a hepta-mannose sugar (with the methyl phosphate, βGlcNAc and UndPP removed). These systems were built using the MemProtMD protocol[61]. Briefly, the protein-sugar complex was placed in a simulation box with POPE lipids orientated randomly around it. In addition to the bonds implicit in the Martini force field, elastic bonds of 1000 kJ mol$^{-1}$ nm$^{-2}$ were applied between protein beads within 1 nm. The systems were then solvated with Martini water and ions to a neutral charge and 0.15 M, before 100 ns of simulation with 20 fs time steps, to allow the POPE bilayer to form around the protein. The substrate was then pulled from the pore, where the sugar-protein distance was increased in the xy directions at a rate of 1 nm per µs. Lipid binding was observed in all three repeats, and binding was quantified by following the distance between the specific lipid and the center of geometry of the residues labelled in Supplementary Fig. 4c. Post 4 µs snapshots were converted back to an atomistic description[62] and simulated as described above.

**Reporting summary.** Further information on experimental design is available in the Nature Research Reporting Summary linked to this article.

## Data availability

Atomic coordinates and structure factors have been deposited in the PDB under accession code 6M96. All other data not already present in the main text or the supplementary information are available from the corresponding author upon reasonable request.

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

## Acknowledgements

We thank Yunchen Bi and Finn Maloney for comments on the manuscript and the staff at AMX beamline of the National Synchrotron Light Source II, a U.S. Department of Energy (DOE) Office of Science User Facility operated for the DOE Office of Science by Brookhaven National Laboratory under Contract No. DE-SC0012704. C.C. was supported by the Cell and Molecular Biology NIH training grant NIH-5T32GM008136 and J.Z. is supported by NIH grant 1R01GM129666. R.C. M.S. and P.S. were funded by Wellcome (208361/Z/17/Z) and the BBSRC (BB/P01948X/1, BB/R002517/1, and BB/S003339/1). Atomistic simulations were carried out on the ARCHER UK National Supercomputing Service (http://www.archer.ac.uk), provided by HECBioSim, the UK High End Computing Consortium for Biomolecular Simulation (hecbiosim.ac.uk), which is supported by the EPSRC (EP/L000253/1).

## Author contributions

C.C. and J.Z. designed all crystallographic experiments. R.C. and P.S. designed and performed all simulation experiments. C.C., J.Z., R.C., P.S. and M.S. contributed to analyzing data and writing the manuscript.

## Additional information

**Competing interests:** The authors declare no competing interests.

