## [Peer Review File · Nature Communications]

Reviewers' comments:

Reviewer #1 (Remarks to the Author):

The manuscript by the Zimmer group presents a new structure of an ABC transporter for O antigens from *Aquifex aeolius*. The current structure is of a Walker B mutant in the ATP-bound state, whereas the previous structure was nucleotide free. Based on the structures, the authors present a possible mechanism for translocation. The new mechanism can account for the topological complexity of O-antigen translocation. The manuscript is based entirely on crystallographic and MD analysis. Support from biochemical or mutagenesis experiments could have strengthened the (interesting) model, but I realize that assays for the protein's activity are difficult to set.

I have a few comments:

1. The cartoon presented in fig 6b is unclear. Why is ATP binding/hydrolysis not indicated?
2. In general: what prevents backflow of the substrate, once ATP is hydrolysed and ADP and Pi are released? In other words: How does ratcheting take place?
3. Page 5: It is highly surprising that Mg is absent. The explanation that the low pH would strip of the Mg is poorly motivated. Does ATP hydrolysis take place at low pH (in the wt protein)? Does the Walker B-mutation lead to Mg stripping?
4. Detergent molecules: LDAO is a "harsh" detergent. Is there any way to assess whether the protein (wt) is still active after purification in this detergent? I'm asking, because the occupation by detergent molecules of a site in a protein structure has in the past led to erroneous interpretation (LeuT S2 site).
5. Page 7, lines 196-7: structure represents a translocation-competent state: I do not understand what the authors mean. If they want to say that the structure is not a detergent artefact, I find the argumentation not strong.

6. Page 8: what was the rationale to use a pure POPE bilayer? Could the MD simulations provide insight in the lipid selectivity of the gating? Headgroup saturation chainlength? Did the authors try crystallization in the presence of added lipid? to compete out the detergent?

Reviewer #2 (Remarks to the Author):

Caffalete et al present the structure of an antigen ABC transporter. Up to now, lipid-linked polysaccharide transport by the ABC transporters lacks the data about the both structure and especially dynamics and mechanism, which hinders the proper understating of this crucial process. This new manuscript uncovers the new high-resolution WzmWzt structure in a new conformation (ATP bound) and is supplemented by MD simulation trying to shed some light on the possible mechanism and signifies a steady progress in the understanding of ABC transporters. Overall, the manuscript is well organized, and both structure and MD simulation are soundly presented. However, there are some questions needed to be addressed before the publication, which are listed below:

1. In lines 350-358 and also in Fig. 6 the authors postulated that UndPP is flipping before the progressive translocation of the antigen. Therefore, obviously the UndPP part is playing a major role in the transport process. Surprisingly, only the sugar part of the substrate has been used in the pulling simulation. The absence of UndPP in the simulation hinders answering some questions: a) how presumably flipped UndPP will react to the pulling of the antigen? b) Upon the full translocation of the antigen out of the channel, lipid molecules occupy the channel, however could it be possible that the UndPP tail competes with them and re-enter into the channel?

2. Based on ref 62 and the MemProtMD protocol, elastic network restraints have been used for the coarse-grained simulation (although, it would have been good to mention it in the method description). This is an important aspect because the pulling simulation would be limited in a crucial way; the presence of EN restraints most likely changes the antigen translocation mechanism and also will affect the response of the channel to the progressive translocation of the substrate by restricting the global changes (especially if the process requires secondary structure changes). It would have been much better to run the pulling simulation in the atomistic setup and in the absence of any restraints.

3. In the method section it has been mentioned that the CG setup was converted back to the atomistic description after the pulling and was simulated further, however through the manuscript there is no data whether this simulation showed anything interesting.
4. The atomistic simulations are rather short and especially in the substrate-bound simulations judging from the RMSD plot of the channel region it seems that two out of three runs are not completely equilibrated (red and orange curves Suppl Fig 3b).
5. There are so many interesting findings in this paper based on the structure and also MD simulation that require biochemical validation. For example, the interesting idea of progressive translocation of substrate points to a much higher stoichiometry between substrate and ATP hydrolysis than other types of ABC transporters. Moreover, there are also several residues identified being crucial for different parts of transport process that need to be tested experimentally.
6. A minor point, it would be better to move Table 1 to the supplementary section.
7. Another point is about the detailed description of methods, which is in my opinion necessary for the reproducibility. It would be better to put all about the methods used in the manuscript rather than referring to another paper in order to make the paper much clearer for readers

We thank the reviewers for their positive evaluation of our manuscript and insightful comments. As detailed below, we addressed all comments in the revised manuscript.

Reviewer #1 (Remarks to the Author):

1. The cartoon presented in Fig. 6b is unclear. Why is ATP binding/hydrolysis not indicated?
> Fig. 6 has been revised to include the apo-state in which WzmWzt's lateral gate is closed. The figure caption has also been revised accordingly.

2. In general: what prevents backflow of the substrate, once ATP is hydrolysed and ADP and Pi are released? In other words: How does ratcheting take place?

> This is a very good question, which we unfortunately cannot answer at this point. In multimeric AAA-ATPases, substrate backsliding is prevented because only individual subunits exert a power-stroke at any given time, hence while one 'hand' resets, others prevent the backsliding of the polymer. For WzmWzt and also likely for SecA-mediated protein translocation, additional interactions of the polymer with the channel either prevent or significantly reduce backsliding, resulting in an, on average, forward movement.

We previously proposed and discuss in this manuscript the model that the initial reorientation of the UndPP head-group occurs spontaneously upon its insertion into the Wzm channel. The membrane potential very likely facilitates this transition and also prevents 'back-flipping' of the negatively charged pyrophosphate group to the cytosolic side. Regarding the following polysaccharide chain, we can only speculate that interactions of the polymer with channel-lining residues are sufficient to prevent or reduce diffusion. We are optimistic that structures of O antigen-associated WzmWzt will provide insights into these mechanistic details.

We modified the text to address these well-justified comments regarding this point.

3. Page 5: It is highly surprising that Mg is absent. The explanation that the low pH would strip of the Mg is poorly motivated. Does ATP hydrolysis take place at low pH (in the wt protein)? Does the Walker B-mutation lead to Mg stripping?

> We currently do not know why Mg²⁺ is not present at the active site. Our density maps are of exceptional quality and sufficient resolution to identify density for a coordinated Mg²⁺, yet we have no evidence suggesting that it is present. It is possible that the introduced Walker B mutation affects Mg²⁺ coordination or perhaps the Mg²⁺ concentration in the crystallization solution was insufficient. We removed the comment suggesting that the low pH of the mother liquor could have affected Mg²⁺ binding.

4. Detergent molecules: LDAO is a "harsh" detergent. Is there any way to assess whether the protein (wt) is still active after purification in this detergent? I'm asking, because the occupation by detergent molecules of a site in a protein structure has in the past led to erroneous interpretation (LeuT S2 site).

> Whether a detergent is harsh or not primarily depends on protein properties. Stable proteins usually tolerate a wide variety of different detergents, including those that are more likely to remove associated lipid molecules, which tends to be destabilizing to many membrane proteins. However, the *Aquifex aeolicus* WzmWzt transporter is a particularly stable protein that exhibits robust ATPase activity, both in an LDAO-solubilized and membrane reconstituted state (published previously: Bi, Y., et al, Nature 2018). ATPase activity is only observed for a properly assembled transporter, the NBDs

on their own show no detectable hydrolytic activity. Thus, we believe that the observed detergent molecules simply cover a hydrophobic channel volume that, in a lipid bilayer, would be occupied by lipid molecules. Additionally, and in response to the comment raised by the reviewer below, we co-crystallized the transporter and soaked WzmWzt crystals with various phospholipids. This, unfortunately, did not result in stable lipid binding, but significantly reduced the observable detergent electron density inside the channel. This suggests to us that the LDAO molecules are in a dynamic equilibrium with hydrophobic molecules in the micelle, which is in agreement with our proposed lipid-diffusion gating model.

5. Page 7, lines 196-7: structure represents a translocation-competent state: I do not understand what the authors mean. If they want to say that the structure is not a detergent artefact, I find the argumentation not strong.

> We rephrased this statement to indicate that our data suggest that the crystallized state represents the WzmWzt conformation during polysaccharide translocation, i.e. no additional channel widening is required to accommodate the polymer.

6. Page 8: what was the rationale to use a pure POPE bilayer? Could the MD simulations provide insight in the lipid selectivity of the gating? Headgroup saturation chainlength? Did the authors try crystallization in the presence of added lipid? to compete out the detergent?

> Reviewer #1 raises a good point, and we did indeed carry out a number of coarse-grained simulations to probe the nature of lipid specificity by the transporter. The data however shows no discernible lipid selectivity with respect to the lipid plug region of the channel, supporting the model that any type of lipid could function as a plug.

We believe that using a PE lipid is a good choice for a bacterial membrane protein, and because we do not expect lipid head-group or acyl chain selectivity with regard to gating, we reasoned that simulations in a pure POPE membrane suffice to reveal the underlying principle.

In an attempt to replace the observed LDAO molecules inside Wzm's periplasmic channel exit with lipid molecules, we co-crystallized the transporter in the presence of POPE and other phospholipids as well as soaked existing crystals with synthetic lipids. Although we cannot identify well ordered lipid molecules inside the periplasmic channel exit, an interesting observation is that in these cases, the detergent density inside the channel is significantly reduced, in some cases even absent. Thus, it is possible that lipids enter the channel and are responsible for disordering the detergent molecules. If requested, we would be happy to provide these density maps to the reviewer.

Additional efforts to crystallize the transporter in a lipid bilayer, i.e. bicelles or HILIDE, are ongoing but have not resulted in promising leads yet.

Reviewer #2 (Remarks to the Author):

1. In lines 350-358 and also in Fig. 6 the authors postulated that UndPP is flipping before the progressive translocation of the antigen. Therefore, obviously the UndPP part is playing a major role in the transport process. Surprisingly, only the sugar part of the substrate has been used in the pulling simulation. The absence of UndPP in the simulation hinders answering some questions: a) how presumably flipped UndPP will react to the pulling of the antigen? b) Upon the full translocation of the antigen out of the channel, lipid molecules occupy the channel, however could it be possible that the UndPP tail competes with them and re-enter into the channel?

> This is a very interesting question that we have not addressed yet. As proposed in our earlier publication (Bi. Y., et al Nature 2018) and discussed in this manuscript, we believe that the UndPP head-group initially inserts into the channel from the cytosolic side and most likely flips spontaneously to the periplasmic side. This event could be facilitated by the negative membrane potential which would also prevent backsliding. Because UndPP is attached to the polysaccharide moiety, this reorientation would also lead to the insertion of the polymer into the channel proper. Because we do not exactly know where and how the UndPP-O antigen molecule binds to the transporter to initiate translocation, we prefer to limit the discussion and speculation on these initial events. After UndPP reorientation and lateral release into the periplasmic leaflet of the bilayer, we believe that it is unlikely that the polyprenyl moiety plays an active role in polysaccharide translocation.

Our simulation studies are intended to address the very last steps of O antigen translocation in which most of the polymer is already translocated. At this time, the UndPP moiety has likely already diffused away from the transporter, such that it would not be affected by the final translocation step. The polysaccharide moiety, however, binds within a well-defined region with a predictable orientation and therefore we deemed it most suitable as the focus for this study.

In principle, it is possible that the UndPP tail competes with lipid molecules for binding to the channel in the absence of a polysaccharide. As such, a partially inserted UndPP tail could also seal the channel, while its complete insertion is highly unlikely considering the length of the polyprenyl chain. In addition, the UndPP-linked O antigen is transferred to lipid A by the WaaL ligase upon release from the ABC transporter and the resulting UndPP is converted to UndP and shuttled back to the cytosolic side. Hence, phospholipids will always be in vast molar excess, making a phospholipid gating mechanism more likely.

2. Based on ref 62 and the MemProtMD protocol, elastic network restraints have been used for the coarse-grained simulation (although, it would have been good to mention it in the method description). This is an important aspect because the pulling simulation would be limited in a crucial way; the presence of EN restraints most likely changes the antigen translocation mechanism and also will affect the response of the channel to the progressive translocation of the substrate by restricting the global changes (especially if the process requires secondary structure changes). It would have been much better to run the pulling simulation in the atomistic setup and in the absence of any restraints.

> This is a good point, however these analyses were designed to test the nature of the interaction of the substrate with the crystal-structure conformation – not to look at structural changes in the protein. This is particularly the case for this experiment, as we were keen to reduce any non-physiological conformational changes from occurring as the substrate is pulled from the transporter; this was therefore achieved with the use of EN restraints. It is important to note, however, that the side chains are fully flexible during the pulling simulation, which will provide a certain degree of conformational flexibility.

Additionally, we have added details regarding the use of EN restraints to the coarse-grained simulation methods section.

3. In the method section it has been mentioned that the CG setup was converted back to the atomistic

description after the pulling and was simulated further, however through the manuscript there is no data whether this simulation showed anything interesting.

> The purpose of these simulations was principally to measure the degree of water flux through the channel. As such, we limited our analyses on the protein's conformational changes.

4. The atomistic simulations are rather short and especially in the substrate-bound simulations judging from the RMSD plot of the channel region it seems that two out of three runs are not completely equilibrated (red and orange curves Supplementary Fig. 4b).

> The length of the simulations was chosen to fully sample the presence of water flux – typically equilibration of channels with water occurs within a few 10s of ns. This suggests that 100 ns will be sufficient for our purpose – especially considering we have run 3 independent repeats of each simulation (providing better sampling than one longer single simulation).

5. There are so many interesting findings in this paper based on the structure and also MD simulation that require biochemical validation. For example, the interesting idea of progressive translocation of substrate points to a much higher stoichiometry between substrate and ATP hydrolysis than other types of ABC transporters. Moreover, there are also several residues identified being crucial for different parts of the transport process that need to be tested experimentally.

> The reviewer is correct, and we acknowledge that an in vitro O antigen transport assay will be an important tool for future studies on the processive translocation mechanism, in particular mutational studies for residues involved in O antigen transport. At this point, however, functional analyses are limited to in vivo secretion assays, which significantly complicates data interpretation. The main obstacle being the multi-step enzymatic synthesis of a lipid-linked O antigen translocation substrate and the unknown chemical structure of the Aquifex aeolicus O antigen. Establishing such a chemically-defined O antigen translocation system is a major ongoing effort.

6. A minor point, it would be better to move Table 1 to the supplementary section.

> We are happy to include Table 1 in either the main text or the supplementary section depending on the preference of the editorial team.

7. Another point is about the detailed description of methods, which is in my opinion necessary for the reproducibility. It would be better to put all about the methods used in the manuscript rather than referring to another paper in order to make the paper much clearer for readers.

> The Methods section has been thoroughly revised to include all missing information.

REVIEWERS' COMMENTS:

Reviewer #1 (Remarks to the Author):

The authors have responded adequately to my concerns

Reviewer #2 (Remarks to the Author):

In the revision, authors have reasonably answered most of the issues raised in the review process, though not marking the changes in the revised manuscript made it very difficult to track them.

I am not still convinced that the UndPP moiety would not influence the final translocation step and in order to realistically understand the translocation process, the progressive translocation should be studied in the presence of the complete ligand. However, I understand that such a study would not be so easy to implement and could be part of a future project.